# Immune Responses to SARS-CoV-2 Variants WT and XBB.1.9: Assessing Vulnerabilities and Preparedness

**DOI:** 10.3390/vaccines13111167

**Published:** 2025-11-16

**Authors:** Limor Kliker, Michal Mandelboim, Menucha Jurkowicz, Neta S. Zuckerman, Enosh Tomer, Yaniv Lustig, Lital Keinan-Boker, Victoria Indenbaum, Ravit Bassal

**Affiliations:** 1The Central Virology laboratory, Ministry of Health, Sheba Medical Center, Ramat Gan 5262, Israel; limor.kliker@sheba.health.gov.il (L.K.); menucha.jurkowicz@sheba.health.gov.il (M.J.); neta.zuckerman@sheba.health.gov.il (N.S.Z.); enosh.tomer@sheba.health.gov.il (E.T.); yaniv.lustig@sheba.health.gov.il (Y.L.); viki.indenbaum@sheba.health.gov.il (V.I.); 2Department of Epidemiology and Preventive Medicine, School of Public Health, Gray Faculty of Medical, Tel Aviv University, Tel Aviv 6997801, Israel; ravit.bassal@moh.health.gov.il; 3The Israel Center for Disease Control, Ministry of Health, Ramat Gan 5262, Israel; lital.keinan2@moh.gov.il; 4School of Public Health, University of Haifa, Haifa 3498838, Israel

**Keywords:** SARS-CoV-2, XBB.1.9, neutralizing antibodies, immune escape, vaccine effectiveness, population immunity

## Abstract

Objectives: The emergence of SARS-CoV-2 variants with enhanced immune evasion capabilities poses ongoing challenges for maintaining population-level immunity. This study aim to evaluate neutralizing antibody responses to the wild-type (WT) strain and the Omicron sublineage XBB.1.9 in the Israeli population using serum samples collected between August 2022 and January 2023, prior to widespread circulation of XBB.1.9. Methods: Pseudovirus-based microneutralization assays incorporating variant-specific spike proteins were employed to measure neutralizing geometric mean titers (GMTs) across subgroups categorized by age, gender, socioeconomic status, and geographic region. Results: Neutralizing titers against XBB.1.9 were significantly lower than those against WT across all demographic groups, with a 29-fold reduction in neutralization activity against XBB.1.9, underscoring the immune escape potential of XBB.1.9. For WT, older adults (≥65 years) exhibited higher titers than younger individuals (*p* < 0.01), whereas no significant age-related differences were observed for XBB.1.9 (*p* > 0.05). Regional disparities in WT immunity were identified, with higher titers in Northern Israel compared to Jerusalem and Southern regions. By contrast, XBB.1.9 neutralization showed no significant regional variation. Conclusions: These findings demonstrate substantially reduced neutralization of XBB.1.9 compared to WT and reveal disparities in WT immunity by age and region. The results emphasize the need for updated vaccines targeting immune-evasive variants and for tailored vaccination strategies to address regional and demographic gaps in protection.

## 1. Introduction

The coronavirus disease 2019 (COVID-19) pandemic, caused by severe acute respiratory syndrome coronavirus 2 (SARS-CoV-2), has led to substantial global morbidity, mortality, and disturbance since its initial detection in December 2019 in Wuhan, China. The virus, which is highly transmissible among humans, causes respiratory illness with symptoms ranging from mild, such as fever, cough, and headache, to severe disease requiring hospitalization and leading to long-term complications or death [1,2,3]. In Israel, the first confirmed COVID-19 case was reported in February 2020 [4].

Since its emergence, SARS-CoV-2 has continuously evolved, giving rise to numerous variants with distinct genetic and phenotypic changes. The World Health Organization (WHO) classified several variants, such as Alpha, Beta, Delta, and Omicron, as Variants of Concern (VOCs), each contributing to distinct waves of infection globally and regionally. The XBB lineage, a recombinant subvariant of Omicron BA.2.75, was first identified globally in August 2022 and has since gained attention due to its increased transmissibility and potential for immune evasion [5]. In Israel, XBB variants began circulating toward the end of 2022, contributing to a surge in COVID-19 cases [6]. Among the XBB lineage, the XBB.1.9 sublineage has emerged as a recombinant variant of particular concern. First identified globally in late 2022, XBB.1.9 is characterized by extensive mutations in the spike protein, including changes in the receptor-binding domain (RBD) that enhance its ability to evade neutralizing antibodies [7]. Comparative genomic analyses have shown that the XBB lineage carries multiple spike protein mutations relative to the WT strain, many of which are located in the RBD. Among these, substitutions such as R346T, N460K, and F486S have been associated with increased resistance to neutralizing antibodies and increased affinity to the ACE2 receptor, thereby promoting immune evasion and higher transmissibility [5,8]. The sublineage XBB.1 acquired an additional mutation, G252V, which is retained in subsequent descendants including XBB.1.9 [9]. detailed genomic comparisons of these mutations across the XBB lineage are summarized in Appendix A.

These mutations contribute to increased transmissibility and immune escape capabilities, challenging existing immunity derived from prior infections or vaccination and raised concerns regarding the population’s preparedness and highlighting the importance of updating vaccine strategies and public health measures.

During December 2020, the Comirnaty (BNT162b2) mRNA vaccine was approved by the US Food and Drug Administration (FDA). This mRNA vaccine was developed against the spike protein of the WT virus. The vaccine was previously shown to have 95% efficacy against the WT and the Alpha variants [10] and had reduced effectiveness against the Delta variant [11]. Omicron sub-lineages were even less affected by the vaccine [12]. In order to improve the protection against circulating Omicron sub-lineages, bivalent COVID-19 booster vaccines were developed [13]. In September 2022, the FDA approved the BNT162b2 bivalent vaccine targeting both WT and Omicron (BA.4/BA.5) variants [14]. Since then, two additional updated vaccines were authorized—a monovalent mRNA vaccine targeting the SARS-CoV-2 Omicron XBB.1.5 [15]. and the monovalent mRNA vaccine targeting KP.2 [16].

Our previous studies have provided important insights into the development of immune responses against novel variants of SARS-CoV-2.

Neutralization assays performed during 2022 showed that breakthrough infections with Omicron BA.1 and BA.5 in vaccinated individuals elicited cross-reactive antibodies; however, these antibodies were significantly less effective against later Omicron subvariants such as BA.4, BA.5, and BA.2.75, reflecting a decrease in the extent of neutralization with subsequent exposure to the variants [17,18]. Recently, administration of a sixth monovalent mRNA vaccine of the XBB.1.5 type in heart transplant recipients elicited robust neutralizing responses not only against XBB.1.5 but also against other Omicron subvariants, highlighting the value of updated monovalent formulations in improving protection in immunocompromised populations [19].

Israel was among the first countries to implement a national vaccination campaign, beginning in December 2020 with the Comirnaty (BNT162b2) vaccine. The campaign quickly expanded to include booster doses as new variants emerged [20].

Emerging variants, such as those in the Omicron lineage, present new challenges due to increased transmissibility and immune evasion. Reduced neutralization has been observed in vaccinated or previously infected individuals [21], although updated boosters, particularly bivalent formulations, provide some protection [22]. Assessing population-level immunity is critical for identifying high-risk groups, informing vaccine strategies, and optimizing public health measures.

This study focuses on assessing population immunity in Israel prior to the wide circulation of the Omicron sublineage XBB.1.9. By analyzing neutralizing antibody titers from serum samples collected prior to its emergence, we aim to evaluate the population’s preparedness and susceptibility to current and future variants, including XBB.1.9 and its sublineages. Understanding population immunity levels provides valuable insights into the likely impact of emerging variants and informs proactive public health strategies.

## 2. Methods

### 2.1. Study Population and Setting

This cross-sectional study analyzed 1140 serum samples collected from all ages from diverse geographical regions within Israel during 1 August 2022–31 January 2023. The samples were collected from laboratories included in the Israel National Sera Bank (INSB) established in 1997 in the Israel Center for Disease Control. For each sample, the data recovered were age, gender, district of residence (North, Central, Jerusalem, South and Judea and Samaria), population group (Jews and others vs. Arabs) and residential socio-economic rank, based on city of residence. Local municipalities in Israel were ranked using an index ranging between 1 (lowest) and 10 (highest) published by the Central Bureau of Statistics, based on a range of variables [23]. The data was accessed for research purposes on 26 February 2023. The INSB sample collection was approved by the legal department of the Israeli Ministry of Health and is completely anonymous. As the samples are anonymized, no additional clinical or immunological data were available beyond the basic sociodemographic information collected (e.g., age, gender, district of residence, population group, and residential socioeconomic rank). Therefore, critical variables such as the time elapsed since vaccination or prior infection, type of vaccine administered, or specific prior infection history were not accessible. Detailed sera collection and methods used for the INSB were previously described [24].

### 2.2. Neutralization Assay

A SARS-CoV-2 lentivirus-based neutralization assay was performed to assess the WT and XBB.1.9, neutralizing antibody levels measured in 50% inhibitory dilution (ID_50_). The neutralizing assay was performed as previously described [6], with minor modifications. Lentiviral particles were produced by co-transfecting HEK293T/17 cells with an expression vector encoding variant-specific SARS-CoV-2 spike alongside packaging vector pCMVDR8.2, luciferase reporter vector pHR′CMV-Luc and a TMPRSS2 expression vector (as part of a partnership with Dr. Daniel Douek, Vaccine Research Institute (VRC), National Institute of Health (NIH), Bethesda, MD, USA). Supernatant was collected from cells 48-h post-transfection and used for subsequent neutralization. Transfection was done using Lipofectamine 3000 (Thermo Scientific, Waltham, MA, USA, cat# L3000001) as specified by the manufacturer. For neutralization, serum samples were heat-inactivated at 56 °C for 30 min. Serum samples were 2-fold diluted in a 96-well plate in dilution medium (MEM 5% FBS), overlaid with pseud-typed Lentivirus solution and incubated at 37 °C for one hour. Pseudovirus–serum complexes were then overlaid with HEK293 TMPRSS2-ACE2 cells suspended in dilution medium. Cells were incubated at 37 °C for 72 h. Following incubation, luminescence was quantified by lysing the cells with tissue culture lysis reagent (Promega, Madison, WI, USA, cat# E1531) and adding luciferase assay substrate (Promega, cat# E1501). Luminescence was read using a Varioskan LUX Multimode Microplate Reader (Thermo Scientific).

### 2.3. Statistical Analysis

Geometric Mean Titers (GMT) and corresponding 95% confidence intervals (CIs) were calculated using GraphPad Prism 10.2.2 (GraphPad Software, Inc., San Diego, CA, USA). Neutralizing antibody titers against the WT and XBB.1.9 SARS-Cov-2 variants were compared using the Wilcoxon signed rank test for paired samples across the following groups: age, gender, nationality, geographic location, and socioeconomic status. Comparisons within variant groups (WT or XBB.1.9) were assessed using the Mann–Whitney test for two-group comparisons and the Kruskal–Wallis test for comparisons across more than two groups. To account for multiple subgroup comparisons, *p*-values were adjusted using the Holm correction to control the family-wise error rate. The geometric mean ratio (GMR) of XBB.1.9 to WT titers was calculated on log-transformed data with the mean of the log differences exponentiated to obtain the overall GMR and corresponding 95% CIs.

## 3. Results

To evaluate population-level immunity and preparedness for the emerging SARS-CoV-2 Omicron sublineage XBB.1.9, we monitored the prevalence of SARS-CoV-2 variants in Israel and analyzed neutralizing antibody titers from 1140 serum samples collected prior to the widespread circulation of XBB.1.9.

The frequency of SARS-CoV-2 variants in Israel was monitored via whole-genome sequencing of positive samples (Israel national consortium for SARS-CoV-2 sequencing, Ministry of Health). Figure 1 presents the monthly distribution of variant prevalence from March 2022 to December 2023. In early 2022, BA.1 and BA.2 were the most prevalent, followed by BA.5, which dominated from mid-2022 to November 2023. Subsequently, BQ and BA.4 gained prominence in early 2023. The XBB lineage, including XBB, XBB.1.5, and especially XBB.1.9, emerged in 2023 and gradually increased in prevalence, dominating Israel’s variant endemicity with 93% between February and July of 2023. XBB.1.9 (in red) appeared between February and July 2023. After April 2023, the detection rate of XBB.1.9 increased, rising from 24% of all sequenced samples in April to 46% in July. Towards the end of 2023, further variation of circulating variants was observed, with a decrease in the prevalence of XBB.1.9 as other lineages emerged.

Next, we evaluated neutralizing antibody titers against WT and XBB.1.9. Serum samples were collected between 1 August 2022 and 31 January 2023, from a cohort of 1140 participants representing the general population. Socio-demographic characteristics of the participants are presented in Table 1.

When comparing neutralizing antibody titers for SARS-CoV-2 WT and XBB.1.9 variants (Figure 2), a statistically significant difference was observed (*p* < 0.0001). Antibody levels of WT (geometric mean titer (GMT) = 1180; 95% CI: 1057–1317) were markedly lower against XBB.1.9 (GMT = 40.69; 95% CI: 36.97–44.77). The GMR of XBB.1.19 to WT titers was 0.034 (95% CI 0.031–0.038), indicating an approximately 29-fold reduction in neutralizing activity against XBB.1.9.

A significant difference in neutralizing antibody titers was observed between age groups when assessing immunity to the WT SARS-CoV-2 variant (Figure 3A), with older individuals (65+) showing higher mean titers compared to younger group (age 30–39), GMT 1257 (CI: 1093–1444) and 1043 (CI: 871.3–1248) respectively (*p* < 0.01). However, no significant differences were detected between age groups when assessing titers against XBB.1.9 (*p* > 0.05).

Region analysis (Figure 3B) revealed further disparities in responses to the WT variant. Participants from the Northern district exhibited the highest GMT (1590, CI: 1321–1912) against the WT variant, followed by participants from the Central region (1203, CI: 922–1569) and Judea and Samaria (1191, CI: 734.4–1930). Lower GMTs were observed among participants from Jerusalem (1075, CI: 825–1400) and the Southern district (911.7, CI: 735.4–1130), with statistically significant differences found between the North and South (*p* < 0.001) and between the North and Jerusalem (*p* < 0.05). No significant differences were detected between other regional pairings. In contrast, when evaluating neutralizing antibody titers against XBB.1.9, GMTs were relatively consistent across regions, ranging between 32.12 in Jerusalem and 53.25 in the Central district, with no statistically significant differences observed (*p* > 0.05).

Across other demographic factors, including gender, socioeconomic status and nationality (Figure 3C–E), no significant differences in neutralizing antibody responses were detected for either the WT or Omicron sublineage XBB.1.9 (*p* > 0.05 for all comparisons).

However, when comparing GMTs between WT and XBB.1.9 within each individual subgroup (e.g., within each age, gender, region, and socioeconomic group), a statistically significant reduction in neutralizing antibody titers was observed (*p* < 0.001 for all comparisons). This consistent intra-group reduction indicates that the Omicron sublineage XBB.1.9 elicited markedly lower neutralization responses relative to the WT variant across all population segments.

## 4. Discussion

The SARS-CoV-2 XBB lineage was first isolated in 2022. The spike (S) protein of the XBB lineage contains 14 mutations in addition to those found in the BA.2 lineage, including nine located specifically within the receptor-binding domain (RBD). The XBB.1 sublineage carries an extra mutation at position G252V compared to the original XBB lineage [5]. XBB.1.9.1 possesses amino acid substitutions in virus proteins that may affect infectivity, viral replication, transmissibility and/or pathogenicity [25].

Israel’s COVID-19 vaccination program has been recognized worldwide for its effectiveness and rapid implementation. The campaign, launched in December 2020, demonstrated the benefits of prioritizing high-risk groups such as healthcare workers and older adults, while utilizing digital infrastructure to monitor vaccine distribution and immune responses efficiently [26]. Booster doses were introduced immediately to address waning immunity and emerging variants, with high compliance significantly reducing morbidity during waves driven by Alpha, Delta, and early Omicron sub-lineages [27,28].

This study aimed to assess antibody responses in the Israeli population prior to the widespread circulation of XBB.1.9, providing a valuable baseline for understanding the population’s preparedness for emerging variants. We evaluated neutralization antibody titers against SARS-CoV-2 variants, WT and XBB.1.9 in the Israeli population focusing on age-related variations, regional differences and demographic disparities. According to our findings, neutralizing antibody titers against XBB.1.9 were significantly lower compared to those against the WT variant. This result is consistent with the enhanced immune escape properties of XBB lineages reported globally. Mutations in the spike protein of XBB.1.9 compromise antibody binding and recognition, reducing its susceptibility to immunity derived from prior infection or vaccination [25], even among populations who received updated bivalent vaccine formulations designed to target the WT and initial Omicron sub-lineages [5,6,13,14,15,29].

Age-related differences in immunity were observed for the WT variant, with older adults (65+) demonstrating higher neutralizing antibody titers compared to younger individuals (30–39). No significant differences were detected for the XBB.1.9 sublineage. This is likely attributable to greater uptake of booster doses in the older population, as earlier vaccination campaigns in Israel prioritized older individuals early in the pandemic [26,30]. Booster doses have been shown to amplify antibody responses and extend the duration of protection [31], which contributed to the stronger immunity observed against the WT variant in this age group. However, the lack of significant differences in antibody titers for XBB.1.9 suggests that immune escape mechanisms of this variant effectively reduced the benefits associated with higher vaccine uptake in older populations.

Regional differences were also observed in this study, particularly in immunity to the WT variant. Individuals from Jerusalem and Southern Israel exhibited significantly lower neutralizing antibody titers against the WT compared to those from Northern Israel. These differences likely reflect disparities in vaccine acceptance rates in public health outreach [32]. Regions such as Jerusalem, which have higher concentrations of ultra-Orthodox Jewish and Arab populations, often face barriers to vaccination rooted in cultural beliefs, socioeconomic factors, and misinformation [33,34]. Similarly, Southern Israel, characterized by geographically dispersed rural communities and larger Bedouin populations, faces systemic inequities in healthcare infrastructure that hinder vaccination coverage [35]. The observed regional differences in neutralizing antibody titers against the WT variant suggest the potential need for targeted vaccination campaigns and outreach programs tailored to culturally diverse and disadvantaged populations.

In contrast, no significant regional differences were observed for neutralizing antibody titers against XBB.1.9, suggesting that the ability of XBB.1.9 to evade the immune system is consistent across populations, regardless of differences in vaccination rates or access to healthcare. Additionally, neutralization levels against XBB.1.9 were consistently lower in all subgroups compared to the WT variant, highlighting an underlying vulnerability across the population.

Recent studies have confirmed that the XBB. lineage continues to evolve with significant immune evasive properties. For example, Uraki et al. demonstrated antiviral efficacy and replicative fitness of a clinical isolate of XBB.1.9.1, showing that it retains susceptibility to several antiviral drugs while resisting neutralization by many monoclonal antibodies [25]. In a genomic surveillance study from Ethiopia, Aga et al. reported high diversity of SARS-CoV-2 lineages among vaccine breakthrough infections in Addis Ababa, highlighting the potential for immune escape in regions with heterogeneous vaccine coverage [36]. Similarly, a large population-based analysis in Nebraska estimated that vaccine effectiveness of XBB.1.5-based boosters wanes over time, achieving ~52% efficacy against infection and ~67% against hospitalization at 4 weeks, with gradual decline thereafter [15]. These results highlight the importance of adapting vaccine formulations and implementing genomic surveillance strategies to monitor emerging sublineages and inform public health responses.

Today, the COVID-19 vaccine landscape continues to evolve in response to the challenges posed by emerging variants such as XBB.1.9. Updated vaccine formulations, including bivalent vaccines targeting both the WT and Omicron sub-lineages (BA.4/BA.5), have already been deployed worldwide and in Israel. In addition, newer monovalent vaccines specifically targeting XBB.1.5 and its recombinant sub-lineages have been approved in several countries, providing a promising avenue for improving immunity in the general population. However, vaccine uptake has declined in recent months due to factors such as pandemic fatigue, misinformation, and reduced risk perception. Given recent declines in vaccine uptake and the need for sustained immunity, periodic integration of COVID-19 vaccines into annual vaccination campaigns may stabilize vaccine compliance and extend protection to the entire population.

The findings of this study reveal critical insights into population-level immunity in Israel against SARS-CoV-2 variants, specifically the wild type (WT) and XBB.1.9. The study highlights the significant immune evasion capabilities of XBB.1.9, characterized by reduced neutralization, as well as notable disparities in immunity across age groups and geographic regions, reflecting the interplay of vaccination coverage, sociodemographic factors, and healthcare access. These results carry important public health implications. First, addressing regional disparities in vaccine uptake and immunity through customized interventions, such as culturally sensitive education campaigns and improved healthcare accessibility in underserved areas, is essential to ensure equitable protection across populations. Additionally, integrating demographic and regional factors into surveillance systems and vaccination strategies can help optimize resource allocation and strengthen overall population immunity against future waves of SARS-CoV-2.

While this study provides valuable insights into immunity prior to the widespread circulation of XBB.1.9, it has several limitations. First, the cross-sectional design does not allow assessment of immune waning over time. Secondly, the study did not include cellular immunity measurements or clinical correlates of protection, such as breakthrough infection or hospitalization data. Future research integrating these parameters is essential to further refine understanding of population preparedness for emerging variants.

## 5. Conclusions

These findings highlight significant gaps in population-level immunity to XBB.1.9 and underscore the importance of updated vaccines and personalized interventions to reduce vulnerabilities. By improving vaccination coverage, enhancing surveillance systems, and designing proactive strategies to address gaps in access to health services, the global health community can better prepare for the health impacts of evolving SARS-CoV-2 variants and strengthen resilience against future pandemics.

## Figures and Tables

**Figure 1 vaccines-13-01167-f001:**
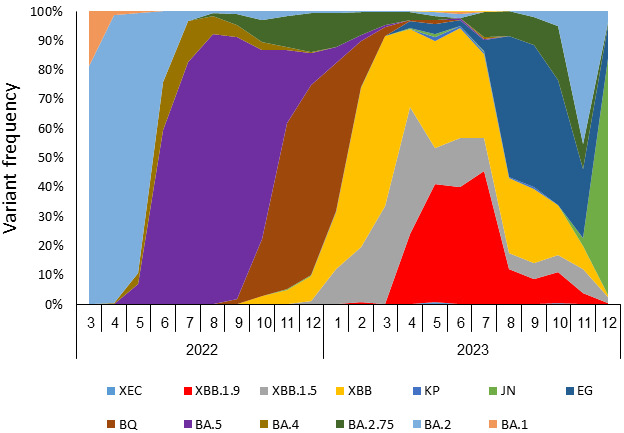
SARS-CoV-2 variant frequencies in Israel, March 2022–December 2023.Frequency of SARS-CoV-2 variants as detected by whole genome sequencing of SARS-CoV-2-positive samples.

**Figure 2 vaccines-13-01167-f002:**
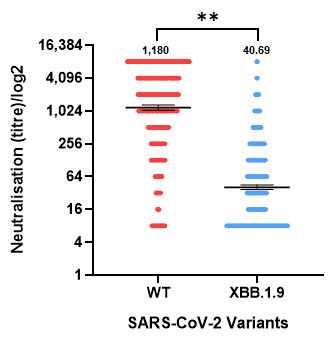
Neutralizing antibody titers against the SARS-CoV-2 wild-type (WT) and XBB.1.9 variants in the population in Israel. Serum samples were tested with neutralization against WT SARS-CoV-2 and XBB.1.9. Geometric mean titers (horizontal lines) with 95% confidence intervals (I bars) are presented, as well as the geometric mean titer value. Dots indicate individual serum samples. A significant reduction in immunity was observed against XBB.1.9. ** *p* ≤ 0.01.

**Figure 3 vaccines-13-01167-f003:**
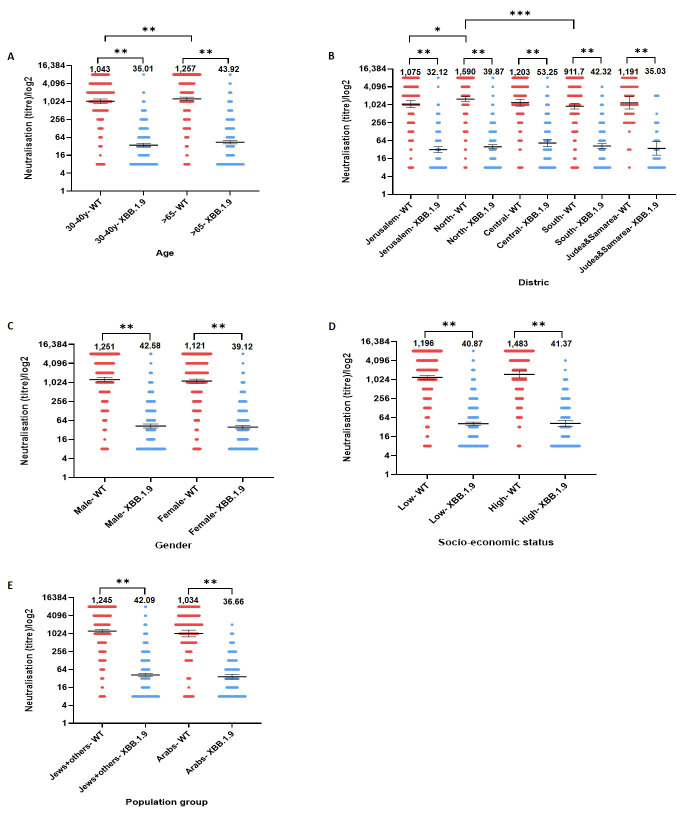
Neutralizing antibody titers against the SARS-CoV-2 wild-type (WT) and XBB.1.9 variants across selected population subgroups. Geometric mean titers (GMT) of neutralizing antibodies against SARS-CoV-2 WT and XBB.1.9 variants are shown by (**A**) age, (**B**) District, (**C**) gender, (**D**) socio-economic status and (**E**) population group. Across all subgroups, neutralization titers were significantly lower for XBB.1.9 compared to WT (*p* < 0.001). Significant differences in GMTs for the WT variant were observed between age groups (higher in adults ≥65 years), and across regions (notably lower in Jerusalem and Southern Israel compared to the North). No statistically significant differences in neutralization titers for XBB.1.9 were observed between subgroups. Bars represent GMTs; error bars indicate 95% confidence intervals. * *p* ≤ 0.05, ** *p* ≤ 0.01, *** *p* ≤ 0.001.

**Table 1 vaccines-13-01167-t001:** Demographics and characteristics of the participants.

	Group	Number of Participants (%)	GMT (WT)	GMT (XBB.1.9)	*p*-Value *
	Total participants	1140 (100)	1180	40.69	<0.001
Gender	Male	529 (46.4)	1251	42.58	<0.001
	Female	611 (53.6)	1121	39.12	<0.001
Age group	Age 30–39	385 (33.77)	1043	35.01	<0.001
	Age 65+	755 (66.23)	1257	43.92	<0.001
Socioeconomic status	Low SES	782 (68.6)	1196	40.87	<0.001
	High SES	189 (16.58)	1483	41.37	<0.001
Population group	Jews and others	855 (75)	1245	42.09	<0.001
	Arabs	214 (18.77)	1034	36.66	<0.001
District	Jerusalem	184 (16.14)	1075	32.12	<0.001
	North	372 (32.63)	1590	39.87	<0.001
	Central	181 (15.88)	1203	53.25	<0.001
	South	352 (30.88)	911.7	42.32	<0.001
	Judea & Samaria	46 (4.04)	1191	35.03	<0.001

* *p*-value was calculated between the variants WT and XBB.1.9 of each group. GMT = geometric mean titer.

## Data Availability

Data supporting the study’s findings are available upon request.

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
