# Peer review of "Vaccines2025, 13(11), 1167;https://doi.org/10.3390/vaccines13111167"

_vaccines, 2025, doi:10.3390/vaccines13111167_

Round 1

Reviewer 1 Report

Comments and Suggestions for Authors

This manuscript presents a seroepidemiological study evaluating neutralising antibody responses to the SARS-CoV-2 wild type (WT) and XBB.1.9 sublineage in the Israeli population between August 2022 and January 2023, prior to the widespread circulation of the XBB.1.9.
The study is well designed, the methodology is appropriate, and the findings carry important public health implications.

Comments.
1. COVID-19 vaccine naming:
In lines 58 and 70, the term “BioNTech-Pfizer vaccine” is used. However, the same manufacturer produces multiple vaccine formulations (e.g., bivalent, XBB monovalent). So, the description should be more specific.
Moreover, in line 70, the expression “Pfizer-BioNTech BNT162b2 mRNA vaccine” is somewhat redundant because:
(1) the manufacturer’s name was already mentioned in the previous sentence, and
(2) “BNT162b2” already specifies the mRNA vaccine.
I recommend using either the commercial name (Comirnaty) or the research code (BNT162b2) consistently instead of solely the manufacturer's name to ensure clarity and professionalism.

2. XBB terminology:
Throughout the manuscript, the authors alternately refer to XBB and XBB.1.9 as a variant, subvariant, and lineage, which creates ambiguity.
According to the PANGO classification and the WHO framework, XBB is a recombinant lineage within the Omicron variant, and XBB.1.9 represents one of its sublineages.
Therefore, it would be more accurate to refer to XBB.1.9 as an Omicron sublineage (XBB.1.9) rather than as a separate variant. For clarity and scientific precision, please use consistent terminology throughout (e.g., “Omicron sublineage XBB.1.9”), unless directly comparing a WHO-designated variant.

3. Unusual p-value reporting:
The Abstract and Results report "p < 0.0023", which is unusually precise and inconsistent with conventional statistical reporting. Such precision may imply a false sense of accuracy and is inconsistent with other reported values (e.g., p < 0.0001).
I recommend rounding p-values consistently throughout the manuscript and following a uniform reporting convention.

4. Table 1 format:
Consider combining the “Number of participants” and “% of total participants” columns into a single column, e.g., “Number of participants (%)”, with entries such as “529 (46.40)”.
This adjustment would make the table more concise without losing the information.

5. Geometric mean titer (GMT) fold change:
It would be helpful to include the GMT fold change between WT and XBB.1.9 in the tables and text, including the abstract.
Reporting fold change values would illustrate the magnitude of reduction in neutralising antibodies from the wild type to the newly emerged sublineage, and provide a clearer sense of immunological impact.

6. Multiple comparison adjustment:
Figure 3 appears to involve multiple subgroup comparisons. Please clarify in the Statistical Analysis section whether any multiple-comparison corrections (e.g., Bonferroni or Holm adjustment) were applied.

7. Inconsistent GMT reporting:
The GMT values reported for total participants differ between the main text/Figure 2 (WT: 1180; XBB.1.9: 40.69) and Table 1 (WT: 1057; XBB.1.9: 36.97).
Please verify and correct this inconsistency to ensure internal accuracy.

Errors.
1. Date format (line 87):
Use full-year notation (2022, 2023) instead of abbreviated forms (22, 23) for consistency with the rest of the text.

2. Subscript notation:
Use the proper subscript format ID₅₀ for the neutralisation titre indicator.

3. Typographical error (line 158):
“Serum samverles” should be corrected to “Serum samples”.

Comments on the Quality of English Language

1. Grammar (line 147):
Replace “participants is presented” with “participants are presented”.

Author Response

Reviewer 1:

This manuscript presents a seroepidemiological study evaluating neutralising antibody responses to the SARS-CoV-2 wild type (WT) and XBB.1.9 sublineage in the Israeli population between August 2022 and January 2023, prior to the widespread circulation of the XBB.1.9.
The study is well designed, the methodology is appropriate, and the findings carry important public health implications.

Comments.
1. COVID-19 vaccine naming:
In lines 58 and 70, the term “BioNTech-Pfizer vaccine” is used. However, the same manufacturer produces multiple vaccine formulations (e.g., bivalent, XBB monovalent). So, the description should be more specific.
Moreover, in line 70, the expression “Pfizer-BioNTech BNT162b2 mRNA vaccine” is somewhat redundant because:
(1) the manufacturer’s name was already mentioned in the previous sentence, and
(2) “BNT162b2” already specifies the mRNA vaccine.
I recommend using either the commercial name (Comirnaty) or the research code (BNT162b2) consistently instead of solely the manufacturer's name to ensure clarity and professionalism.

Response: In the revised version, we have specified the vaccine names more precisely to distinguish between the original Comirnaty (BNT162b2) formulation, the bivalent (WT/BA.4–5) vaccine, and the updated monovalent (XBB.1.5 and KP.2) versions.

  1. XBB terminology:
    Throughout the manuscript, the authors alternately refer to XBB and XBB.1.9 as a variant, subvariant, and lineage, which creates ambiguity.
    According to the PANGO classification and the WHO framework, XBB is a recombinant lineage within the Omicron variant, and XBB.1.9 represents one of its sublineages.
    Therefore, it would be more accurate to refer to XBB.1.9 as an Omicron sublineage (XBB.1.9) rather than as a separate variant. For clarity and scientific precision, please use consistent terminology throughout (e.g., “Omicron sublineage XBB.1.9”), unless directly comparing a WHO-designated variant.

Response: We have revised the terminology throughout the manuscript to align with the PANGO and WHO classification, consistently referring to XBB.1.9 as the Omicron sublineage XBB.1.9 instead of a variant and XBB as XBB lineage.

  1. Unusual p-value reporting:
    The Abstract and Results report "p < 0.0023", which is unusually precise and inconsistent with conventional statistical reporting. Such precision may imply a false sense of accuracy and is inconsistent with other reported values (e.g., p < 0.0001).
    I recommend rounding p-values consistently throughout the manuscript and following a uniform reporting convention.

Response: All p-values have been rounded and standardized (e.g., p<0.05, p<0.01, p<0.001) to ensure consistency and adherence to conventional statistical reporting guidelines.

  1. Table 1 format:
    Consider combining the “Number of participants” and “% of total participants” columns into a single column, e.g., “Number of participants (%)”, with entries such as “529 (46.40)”.
    This adjustment would make the table more concise without losing the information.

Response: The columns have been combined to a single column.

  1. Geometric mean titer (GMT) fold change:
    It would be helpful to include the GMT fold change between WT and XBB.1.9 in the tables and text, including the abstract.
    Reporting fold change values would illustrate the magnitude of reduction in neutralising antibodies from the wild type to the newly emerged sublineage, and provide a clearer sense of immunological impact.

Response: We calculated the GMT fold change between WT and XBB.1.9 and added details to the abstract, methods and results.

  1. Multiple comparison adjustment:
    Figure 3 appears to involve multiple subgroup comparisons. Please clarify in the Statistical Analysis section whether any multiple-comparison corrections (e.g., Bonferroni or Holm adjustment) were applied.

Response: We adjusted the p-values using the Holm correction, and updated the p-values that appear in Figure 3, and in the results section. We revised the statistical analysis section to reflect this.

  1. Inconsistent GMT reporting:
    The GMT values reported for total participants differ between the main text/Figure 2 (WT: 1180; XBB.1.9: 40.69) and Table 1 (WT: 1057; XBB.1.9: 36.97).
    Please verify and correct this inconsistency to ensure internal accuracy.

Response: We have verified the data and corrected the GMT values in the main text, Figure 2, and Table 1 to ensure consistency and accuracy throughout the manuscript.

Errors.
1. Date format (line 87):
Use full-year notation (2022, 2023) instead of abbreviated forms (22, 23) for consistency with the rest of the text.

Response: Corrected.

  1. Subscript notation:
    Use the proper subscript format ID₅₀ for the neutralisation titre indicator.

Response: Corrected.

  1. Typographical error (line 158):
    “Serum samverles” should be corrected to “Serum samples”.

Response: Corrected.

Comments on the Quality of English Language.

  1. Grammar (line 147):
    Replace “participants is presented” with “participants are presented”.

Response: Corrected.

Reviewer 2 Report

Comments and Suggestions for Authors

I do not have comments.

Author Response

Reviewer 2

Reviewer 2 had no comments

Reviewer 3 Report

Comments and Suggestions for Authors

Author Response

Reviewer 3:

The authors utilized pseudovirus-based neutralization assays to evaluate the neutralizing capacity of sera from the Israeli population against both the wild-type (WT) SARS-CoV-2 strain and the XBB.1.9 variant. A total of 1,140 serum samples were analyzed, all collected prior to the widespread circulation of the XBB.1.9 and XBB.1.5 variants. The results were further categorized according to demographic and regional factors, including age, sex, socioeconomic status, and geographic location.

The findings demonstrated a significantly reduced neutralization capability against XBB.1.9 across all subgroups, suggesting this variant's potential to evade pre-existing immunity. Although the study adopts a conceptually incremental approach, the inclusion of real-world data from 1,140 serum samples substantially enhances its robustness and relevance. The authors highlight the importance of the XBB lineage; however, the assessment was limited to sera neutralization of XBB.1.9 and did not include XBB.1.6 or XBB.1.5—an important omission given that vaccines were specifically designed for this variant.

Overall, the study provides valuable insights into the immune responses targeting the specific XBB.1.9 variant, making it a significant contribution to COVID-19 research. I believe the article deserves publication after making some substantial revisions.

Authors should address the following questions:

  1. Although similar, authors should specify which variant was investigated in this study: XXB.1.9.1 or XXB.1.9.2. Were both investigated, or only one? Why? Additionally, describe the similarities and differences in sequence, epidemiology, etc. Could we perhaps use Table 1 Including other It depends on the author's preferences.

Response: We specifically investigated the XBB.1.9.1 subvariant.

Identification of SARS-CoV-2 variants from whole-genome sequences is conducted at Israel’s National Virology Laboratory using a mutation table that lists characteristic mutations for each variant. This table was developed by Israel’s Ministry of Health based on GISAID sequence data for the various lineages. Accordingly, the main distinction between XBB.1.9.1 and XBB.1.9.2 involves three amino acid substitutions located outside the spike protein, which are unlikely to impact epidemiological behavior. Synonymous mutations identified in the NSP7, NSP13, and ORF7a genes of SARS-CoV-2 XBB.1.9.1 are unlikely to influence viral transmissibility or immune evasion. Previous studies demonstrated that these non-structural and accessory proteins are functionally conserved across variants, and synonymous substitutions generally do not alter protein structure or replication efficiency [1–3]. Moreover, while NSP13 is involved in viral helicase and replication activities, most sequence changes reported in Omicron lineages, including those outside the spike region, have shown minimal impact on viral fitness [4,5].

Similarly, ORF7a mutations, particularly synonymous ones, have not been associated with significant epidemiological shifts or immune escape [6].

source

aa_sub

genetic_region

mutation

mutation_type

nuc_sub

position

refference

XBB.1.9.1

D38D

NSP7

T

synonymous

C11956T

11956

C

XBB.1.9.2

G38G

ORF7a

C

synonymous

A27507C

27507

A

XBB.1.9.2

T214T

NSP13

T

synonymous

A16878T

16878

A

  1. Harvey WT, Carabelli AM, Jackson B, Gupta RK, Thomson EC, Harrison EM, et al. SARS-CoV-2 variants, spike mutations and immune escape. Nat Rev Microbiol. 2021;19(7):409–424. doi: 10.1038/s41579-021-00573-0
  2. Saldivar-Espinoza B, Moreno-Sandoval H, Utrilla-Trigo S, Fajardo V, García-Machorro J, Cruz-Rangel S, et al. The Mutational Landscape of SARS-CoV-2. Viruses (Basel). 2021;13(10):1921. doi: 10.3390/v13101921
  3. Fumagalli M, Pozzoli U, Cagliani R, Comandatore F, Sironi M. Analysis of 3.5 million SARS-CoV-2 sequences reveals strong purifying selection at synonymous sites. Virol J. 2023;20(1):87. doi: 10.1186/s12985-023-02016-1
  4. Inniss NL, Morris GM, Tucker SJ, Dawson RJK, Hill J, Malla TR, et al. Activity and inhibition of the SARS-CoV-2 Omicron nsp13 R392C variant. SLAS Discov. 2024;29(2):165–176. doi: 10.1016/j.slasd.2023.09.005
  5. Peng Q, Peng R, Yuan B, Zhao J, Wang M, Wang X, et al. Structural and Biochemical Characterization of the SARS-CoV-2 Polymerase Complex. Cell. 2020;182(2):417–428.e13. doi: 10.1016/j.cell.2020.05.031
  6. Arshad N, Cresswell P. SARS-CoV-2 accessory proteins ORF7a and ORF3a use distinct mechanisms to downregulate MHC-I surface expression. Proc Natl Acad Sci U S A. 2022;119(9):e2116785119. doi: 10.1073/pnas.2116785119

  1. Explain clearly why the rest of the lineage was not studied (e.g., 1.6 or XBB.1.5).

Response: The focus of our study was specifically on the XBB.1.9 subvariant due to its emergence in Israel during the study period and its significant increase in prevalence, as documented through whole genome sequencing. While other XBB subvariants, such as XBB.1.6 and XBB.1.5, are certainly relevant to a broader understanding of the immune escape dynamics of SARS-CoV-2, they were not investigated here because their prevalence in Israel during the study period was limited compared to XBB.1.9, which dominated the local variant landscape.

In the context of the XBB lineage, the immune escape capabilities and clinical significance of XBB.1.5 have been investigated extensively in prior publications:

  1. Peled, Yael et al. 2024. “Sixth Monovalent XBB.1.5 Vaccine Elicits Robust Immune Response against Emerging SARS-CoV-2 Variants in Heart Transplant Recipients.” The Journal of Heart and Lung Transplantation 43(7): 1188–92. https://linkinghub.elsevier.com/retrieve/pii/S1053249824015377.
  2. Lustig, Yaniv, Michal Canetti, et al. 2024. “SARS-CoV-2 IgG Levels as Predictors of XBB Variant Neutralization, Israel, 2022­ and 2023.” Emerging Infectious Diseases 30(5). https://wwwnc.cdc.gov/eid/article/30/5/23-1739_article.
  3. Lustig, Yaniv, Noam Barda, et al. 2024. “Humoral Response Superiority of the Monovalent XBB.1.5 over the Bivalent BA.1 and BA.5 MRNA COVID-19 Vaccines.” Vaccine 42(22): 126010. https://linkinghub.elsevier.com/retrieve/pii/S0264410X24006352.

  1. The article describes mutations in the spike proteins of the XXB lineage: a multiple alignment of these variants with the Wuhan wild type should be demonstrated and discussed with its epidemiological This can be presented as a primary or supplementary figure.

Response:  We have added a supplementary table summarizing the differences between the XBB lineage and the Wuhan reference sequence across the genome. This table was generated based on a consensus of GISAID sequences designated as XBB and their comparison to the Wuhan reference. We believe this provides a more accurate representation of the defining differences, as it highlights lineage-specific mutations while excluding patient-specific variations. Furthermore, we have expanded the Introduction section to include a more detailed description of the spike protein mutations characteristic of the XBB lineage and its subvariants, together with a discussion of their epidemiological relevance, supported by recent references.

The references have been placed in brackets, as we were unable to update their numbering in the file.

  1. The selection criteria are not clearly There are no details regarding sample selection, inclusion or exclusion criteria, or the time elapsed since vaccination or infection. These factors should be specified, as they impact the results. If no criteria were applied, please mention this limitation in the materials and methods section of the study.

Response: The serum samples used in this study were obtained from the Israel National Serum Bank (INSB). These serum samples are regularly collected from the general population in Israel and are maintained fully anonymized in compliance with legal and ethical standards. As such, no additional clinical data were available beyond the basic socio-demographic information that we specifically selected for this study, including the age, gender, district, population group, and socio-economic rank. Details regarding the INSB sample collection are described in methods section.

No specific inclusion or exclusion criteria, such as the time elapsed since vaccination or infection, were applied in the study, as this information is unavailable for anonymized samples. We added this detail as a limitation in the Materials and Methods section.

  1. Discussion could more critically compare findings to international

Response: We have expanded the Discussion section to include comparisons with international studies on similar variants, including XBB.1.5. For instance, we now reference findings from global studies that highlight the immune-escape properties of XBB lineages, consistent with the reduced neutralization titers observed in our study for XBB.1.9.

  1. Since this study used pseudovirus-based neutralization assays to evaluate serum samples from COVID-19 patients, and no further experimental results were obtained, it is important to exercise caution when interpreting the The discussion should be based on the results section; therefore, minimizing speculation will improve the reliability of the article.

Response: We have refined the Discussion section to ensure that all statements are directly supported by our results. Speculative content concerning future public health implications has been minimized or fully excluded unless directly grounded in data presented. We have emphasized the need for further studies to validate the findings.

  1. The legend for Figure 3 is divided by letters, but this division is not indicated in the figure itself.

Response: Corrected.